# A Novel Demulsifier with Strong Hydrogen Bonding for Effective Breaking of Water-in-Heavy Oil Emulsions

**DOI:** 10.3390/ijms241914805

**Published:** 2023-09-30

**Authors:** Xiao Xia, Jun Ma, Fei Liu, Haifeng Cong, Xingang Li

**Affiliations:** 1Department of Chemical Engineering, School of Chemistry and Chemical Engineering, Guizhou University, Guiyang 550025, China; xiaxiao2021@163.com (X.X.); ce.feiliu@gzu.edu.cn (F.L.); 2Guizhou Key Laboratory for Green Chemical and Clean Energy Technology, Guiyang 550025, China; 3School of Chemical Engineering and Technology, Tianjin University, Tianjin 300072, China; conghaifeng@tju.edu.cn (H.C.); lxg@tju.edu.cn (X.L.); 4Zhejiang Institute of Tianjin University, Ningbo 315201, China

**Keywords:** demulsifier, hydrogen bonds, interactions, water-in-heavy oil emulsions

## Abstract

In the heavy petroleum industry, the development of efficient demulsifiers for the effective breaking of interfacially active asphaltenes (IAA)-stabilized water-in-heavy oil (W/HO) emulsions is a highly attractive but challenging goal. Herein, a novel nitrogen and oxygen containing demulsifier (JXGZ) with strong hydrogen bonding has been successfully synthesized through combining esterification, polymerization and amidation. Bottle tests indicated that JXGZ is effectual in quickly demulsifying the IAA-stabilized W/HO emulsions; complete dehydration (100%) to the emulsions could be achieved in 4 min at 55 °C using 400 ppm of JXGZ. In addition, the effects of demulsifier concentration, temperature and time on the demulsification performance of JXGZ are systematically analyzed. Demulsification mechanisms reveal that the excellent demulsification performance of JXGZ is attributed to the strong hydrogen bonding between JXGZ and water molecules (dual swords synergistic effect under hydrogen bond reconstruction). The interaction of the “dual swords synergistic effect” generated by two types of hydrogen bonds can quickly break the non-covalent interaction force (π-π stacking, Van der Waals force, hydrogen bonds) of IAA at the heavy oil–water interface, quickly promote the aggregation and coalescence of water molecules and finally achieve the demulsification of W/HO emulsions. These findings indicate that the JXGZ demulsifier shows engineering application prospects in the demulsification of heavy oil–water emulsions, and this work provides the key information for developing more efficient chemical demulsifiers suitable for large-scale industrial applications.

## 1. Introduction

There have been wide concerns around the different types of emulsions involved in the petroleum and petrochemical industries [1,2,3]. Such concerns are particularly present in the industrial processes of heavy petroleum, since the presence of asphaltenes, resin and solid fines are natural surface-active agents in heavy oil [4,5], and so the stable water-in-heavy oil (W/HO) emulsions will be inevitably formed [6]. These stable emulsions will bring a series of problems, such as damage to downstream pipelines and plant equipment, environmental pollution and the reduction of heavy oil quality [7,8]. Therefore, how to effectively realize the demulsification of W/HO emulsions has become a technical bottleneck in the field of heavy petroleum exploitation and processing.

Naturally, clearing the stable mechanism of W/HO emulsions is the prerequisite for breaking emulsions. As is widely known, many research studies have indicated that asphaltenes are the key components to stabilize the W/HO emulsions [9,10,11,12,13]. However, some research has suggested that only a small portion of the asphaltenes in the heavy oil, called interfacially active asphaltenes (IAA), are the main stabilizer to stabilize W/HO emulsions [14,15,16]. Our previous work found that IAA in heavy oil performs well in stabilizing the W/HO emulsions (no demulsification for more than 1 year) [17]. This is mainly ascribed to the IAA which could accumulate at the heavy oil–water interface and self-aggregate to form a viscoelastic interfacial film through π-π stacking, hydrogen bonds and other non-covalent bonds [18]. These non-covalent interactions make IAA strongly adsorbed on the surface of water molecules in the form of viscoelastic interfacial films, and further hindered the coalescence of droplets, resulting in the high stability of W/HO emulsions [19]. Therefore, the core point of the demulsification of W/HO emulsions is how to efficiently disrupt the non-covalent interactions of IAA at the oil–water interface.

As an economically efficient method, chemical demulsification has been widely used to break W/HO emulsions in the heavy oil industry [20,21,22,23]. The key to chemical demulsification lies in demulsifiers [24,25]. At present, there are many different chemical demulsifiers being proposed, such as ionic demulsifiers [26], ionic liquids demulsifier [27], nonionic demulsifiers [28,29], magnetic demulsifier [30], etc. However, our previous research results have suggested that chemical demulsification is an effective method to ruin IAA films [31,32]. Briefly summarizing the research results, in order to break the IAA-stabilized W/HO emulsions, a modified aliphatic alcohol nonionic polyether (MJTJU-2) has been synthesized by introducing some oxygen-containing groups (hydroxyl, ester groups, carboxyl groups and ether groups) [31]. The results showed that this kind of demulsifier could replace the IAA films and break the emulsions by the hydrogen bond reconstruction of different oxygen atoms (Figure 1). These findings implied that constructing the novel demulsifier with strong hydrogen bonding could help to effectively destroy the non-covalent forces of IAA at the oil–water interface. Relevant studies showed that the nitrogen atom is also conducive to forming hydrogen bonds with hydrogen in water molecules [33,34,35,36]. In summary, combined with previous research results, building a novel demulsifier simultaneously containing N and O with strong hydrogen bonding is expected to rapidly destroy the IAA films, and fast demulsifying IAA stabilized W/HO emulsions.

Therefore, herein, the research objectives of this work are to: (i) synthesize a novel demulsifier (JXGZ) simultaneously containing N and O with strong hydrogen bonding through combining esterification, polymerization and amidation, (ii) detect the demulsification performance of JXGZ in breaking IAA-stabilized W/HO emulsions by bottle tests and (iii) unveiling the dual swords synergistic effect of JXGZ under hydrogen bond reconstruction in the demulsification of IAA-stabilized W/HO emulsions by interface characterization and molecular dynamics simulation.

## 2. Results and Discussions

### 2.1. Characterization and Analysis of Demulsifiers

The molecular weight results of FAP, JXGZ and h-PAMAM are displayed in Table 1 and Appendix A. Although the weight average molecular weight (Mw, 10896 g/mol) and number average molecular weight (Mn, 8296 g/mol) of h-PAMAM are higher than FAP (Mw: 4621 g/mol, Mn: 3440 g/ mol) and JXGZ (Mw: 6504 g/mol, Mn, 5620), the relative molecular weight distribution (Mw/Mn) of JXGZ (1.16) is lower than that of FAP (1.39) and h-PAMAM (1.31). It indicates that the molecular weight distribution of JXGZ is the most uniform.

The FTIR spectra of FAP, JXGZ and h-PAMAM are shown in Figure 2a. In the FTIR spectrum of FAP, 3400 cm^−1^ is the absorption peak of hydroxyl group (–OH) [28]. The antisymmetric stretching vibration of –CH_3_ is at 2900 cm^−1^. The absorption peaks of asymmetric deformation vibration and symmetric deformation vibration of -CH_3_ are at 1450 cm^−1^ and 1370 cm^−1^, respectively. In the FTIR spectrum of h-PAMAM, the absorption peak of 2931 cm^−1^ and 2835 cm^−1^ are the asymmetric and symmetric stretching vibrations of -CH_2_, respectively. The typical absorption peak of amido groups (RCONH_2_) appears at 1640 cm^−1^; the stronger absorption band of 3284 cm^−1^ is the amino groups [37]. However, in the FTIR spectrum of JXGZ, 1668 cm^−1^ is the absorption peak of the amido groups; the peak at 3284 cm^−1^ may be an overlap of the amino and hydroxyl groups [38]. The stretching vibration band of 1720 cm^−1^ may be an overlap of ester group (–COOH) and carboxyl (–COOR). In any case, the stretching vibration of 1090 cm^−1^ is ether bond (C–O–C) in FAP and JXGZ.

Figure 2b shows the representative ^13^C NMR spectra of different demulsifiers. The chemical shift of the ester group is between 160 ppm and 170 ppm. The chemical shift of carboxyl is between 171 ppm and 180 ppm [31]. It shows that the JXGZ includes ester and carboxyl groups, and the ^1^H NMR spectra in Figure 2c also confirms this result. In addition, the chemical shift from 2.20 to 2.40 ppm represents NH_2_, and NH in JXGZ and h-PAMAM. The chemical shift from 2.5 to 3.0 ppm represents COCH_2_, CH_2_NH, NH(CH_2_)_2_NH and NH(CH_2_)_2_, and the chemical shift between 3.20 and 3.50 ppm is NCH_2_ [38]. Figure 2c shows a single peak approximately at 4.0 ppm, which is attributed to the hydroxyl group of JXGZ and FAP. Anyway, FAP and JXGZ contain EO (ethylene oxide)-PO (Propylene oxide) chain segments [28]. In Figure 2b, the signal at the 73.4 ppm chemical shift contains two peaks. They are signals of the same and isomorphic main bodies of methylene carbon in the PO binary group (peak area ratio of 1:1). The signal with a chemical shift of 75.5 ppm contains three peaks, which are the same, heterogeneous and inter-host isomorphic signals of PO three unit methylene carbon (peak area ratio 1:2:1). The above analysis results indicate that the JXGZ demulsifier including different polar hydrophilic groups (hydroxyl group, amine groups, ester group, carboxyl group and ether bond) is successfully obtained. Thereinto, the structural schematics of FAP, JXGZ and h-PAMAM are shown in Appendix A.

Based on the requirements of the petroleum industry, demulsifiers used in the petroleum industry should possess a certain thermal stability at application temperatures [39]. Here, Figure 2d shows the thermal decomposition process of FAP, h-PAMAM and JXGZ demulsifiers. It can be observed that the weight loss rates of FAP, h-PAMAM and JXGZ from 25 °C to 60 °C are 0.09%, 0.50% and 0.07%, respectively. The temperature continuously increased from 60 °C to 350 °C, and the weight loss rates of FAP and JXGZ were 14.62% and 4.12%, respectively. In addition, continuously increasing the temperature from 60 °C to 120 °C results in a weight loss of 3.35% for h-PAMAM. Continuously heating the sample above 350 °C will cause the decomposition of organic compounds in FAP and JXGZ. These results indicate that the thermal stability of JXGZ is better than that of FAP and h-PAMAM.

### 2.2. Analysis of Interfacial Activity of Demulsifiers

The interfacial activity of chemical demulsifiers plays an important role in demulsification. Therefore, proving that JXGZ possesses higher interfacial activity is crucial for the subsequent demulsification of W/HO emulsions. The surface tension and interfacial tension related to demulsifiers are displayed in Figure 3. Obviously, the surface tension of pure water can be reduced significantly by both FAP, h-PAMAM and JXGZ. In addition, when the concentration of different demulsifiers exceeds 400 ppm, the surface tension remains constant (Figure 3a). It shows that the critical micelle concentrations (CMC) of FAP, h-PAMAM and JXGZ are 400 ppm. However, the surface tension of the JXGZ aqueous solution is smaller than that of the FAP and h-PAMAM solutions at the same concentration, suggesting that JXGZ possesses the strongest surface activity.

Figure 3b shows that both FAP, h-PAMAM and JXGZ could effectively reduce the oil–water interfacial tension. When the concentration of the FAP, h-PAMAM and JXGZ demulsifiers is 100 ppm, the interfacial tension between oil and water sharply decreases from 36.936 mN/m to 18.325 mN/m, 29.637 mN/m and 14.666 mN/m, respectively. When JXGZ with a concentration of 400 ppm is added, the interfacial tension between oil and water can be reduced to 2.621 mN/m (Figure 3b). Comparing with FAP, h-PAMAM, and IAA, JXGZ performs better in reducing the oil–water interfacial tension (Figure 3c). This further indicates that JXGZ possesses stronger interfacial activity than FAP, h-PAMAM and IAA.

### 2.3. Demulsification Performance of JXGZ

In order to test the demulsification performance of JXGZ in the demulsification of IAA-stabilized W/HO emulsions, the impact of demulsifier concentration, temperature and time on the demulsification performance was explored (Figure 4).

Figure 4a shows the change of the dehydration ratio with the demulsifier concentration. It was found that the W/HO emulsions cannot be demulsified without the demulsifiers. After adding demulsifier into the emulsions, it is observed that the W/HO emulsions can be demulsified, but the demulsification effect depends on the concentration and type of demulsifiers. Thereinto, at 100 ppm, the dehydration rates of FAP, h-PAMAM and JXGZ are 0%, 0% and 40%, respectively. When the demulsifier concentration reaches the CMC, the dehydration effect of the demulsifiers is optimal. The maximum dehydration rates of FAP, h-PAMAM and JXGZ are found to be 20%, 19% and 100%, respectively. The results also show that the JXGZ only takes 4 min to demulsify the emulsions, which is much faster than FAP and h-PAMAM. In addition, the above results show that when the concentration of the demulsifier is CMC of the demulsifier, the dehydration effect of the emulsions is the best. However, when the concentration of the JXGZ demulsifier exceeds CMC, the dehydration rate will remain at 100%. It also indicates that the demulsification performance of JXGZ in IAA-stabilized W/HO emulsions is better than that of FAP and h-PAMAM.

Figure 4b shows the effect of temperature on the demulsification efficiency of demulsifiers. When the demulsification time is 4 min, the temperature increases from 30 °C to 55 °C; when the concentration of JXGZ is 400 ppm, the water removal rate increases from 40% to 100%. However, in the same temperature range, the water removal rate will only increase 20% for FAP and the water removal ratio only increases 19% for h-PAMAM. When the temperature increases up to 70 °C, the water removal ratio reaches 32% and 25% by the FAP and h-PAMAM demulsifiers, respectively. The results indicate that JXGZ possesses a better demulsification performance than FAP and h-PAMAM. From the perspective of molecular thermodynamic movement, the mechanical strength of IAA interfacial films may decrease with the increase of temperature, thus reducing the stability of emulsions. Moreover, the movement of demulsifier molecules would become faster with the increase of temperature [40]. It can accelerate the demulsifier molecules to adsorb the heavy oil–water interface to destroy IAA films.

Figure 4c shows the change to the dehydration ratio of the emulsions with settling time. It was found that the demulsification efficiency of JXGZ in the demulsification of emulsions is more than that of FAP and h-PAMAM. The emulsions could be quickly separated by JXGZ (the dehydration ratio reached 100%) at the 4th min. However, when the settling time exceeds 10 min, the dehydration ratio of FAP and h-PAMAMis only 25% and 23%. Due to the introduction of various hydrophilic groups (ester group, carboxyl group and amino groups), the JXGZ performs better and faster than FAP and h-PAMAM in demulsifying the IAA-stabilized emulsions (Figure 4d). This finding indicates that the different hydrophilic groups in JXGZ play the key roles in demulsifying W/HO emulsions. Figure 5 shows the process photos of emulsions demulsification and the stereomicroscopic photographs of IAA-stabilized W/HO emulsions at 55 °C and different times by using the JXGZ demulsifier with a concentration of 400ppm.

These results indicate that JXGZ possesses a better demulsification effect than reported demulsifiers [17,31,41,42] in demulsifying the asphaltene-stabilized heavy oil–water emulsions. In addition, the oxygen-containing demulsifier (MJTJU-2) we previously reported can realize the demulsification of IAA emulsions in 25 min at 60 °C [31]. After introducing hydrophilic nitrogen-containing groups into MJTJU-2, the demulsification time (4 min) and temperature (55 °C) is significantly reduced, which indirectly indicates that synergistic effect of hydrophilic groups containing oxygen and nitrogen in the demulsification of IAA-stabilized W/HO emulsions.

### 2.4. Analysis of Demulsification Mechanism of JXGZ

In order to further explore the demulsification mechanism of JXGZ in the demulsification process of IAA-stabilized W/HO emulsions, the demulsification mechanism of JXGZ was studied through mesoscopic molecular dynamics and all-atom molecular dynamics simulation calculations.

Figure 6 shows the mesoscopic molecular dynamics simulation results. It was found that the JXGZ demulsifier could promote flocculation and the coalescence of water molecules. After the demulsification of emulsions, the JXGZ and IAA molecules gather together, while IAA molecules are mainly distributed in the oil phase (Figure 6b,c). Most demulsifier molecules are distributed in the aqueous phase, with only a small portion distributed in the oil phase (Figure 6d,e), indicating that JXGZ possesses amphipathy. Figure 7a shows the relative concentration change of water in the demulsification process. At the initial state, water molecules are uniformly dispersed in emulsions system. After the calculation of the mesoscopic dynamic demulsification process is completed, water molecules eventually coalesce on the XOY plane. It shows that water molecules gather together during the dynamic demulsification process. The above results show that JXGZ is conducive to rapidly promoting the aggregation and coalescence of water molecules in emulsions, thus realizing the demulsification of emulsions. Figure 7b shows a radial distribution function (RDF) of IAA, JXGZ and water. The RDF of IAA and water, JXGZ and water and JXGZ and IAA in the demulsification process displayed sharp peaks in the 6.5 Å. It suggests the existence of interactions between JXGZ, water and IAA molecules. In addition, the peak between the JXGZ and water molecule is sharper than that between IAA and water molecule, indicating that there is competition between JXGZ and IAA in the demulsification process of lotion. Based on the RDF, the change of mean force potential (PMF) with distance between the IAA, JXGZ demulsifier and water were calculated (Figure 7c). The interaction energies between JXGZ and water, IAA and water and JXGZ and IAA are −9.6 kJ/mol, −8.6 kJ/mol and −5.5 kJ/mol, respectively. It shows that compared to the interaction between the IAA and water molecules, the interaction between the JXGZ and water molecules is stronger.

Figure 8 shows the results of the all-atom molecular dynamics simulation calculation. After 500 ps of the all-atom molecular dynamics simulation calculation, the IAA molecules can diffuse from the oil phase into the oil–water interface (Figure 8a). However, when the IAA and JXGZ molecules are present simultaneously in the oil phase, JXGZ can preferentially adsorb on the surface of water molecules (Figure 8b). This indicates that the interaction between the JXGZ and water molecules is stronger, which is consistent with the above RDF results. When IAA adsorbed to the oil–water interface, JXGZ is added to the oil phase to continue the all-atom molecular dynamics calculation. It was found that JXGZ can replace IAA from the surface of the water molecule (Figure 8c). This is due to JXGZ possessing a higher diffusion coefficient (JXGZ: 38.55 × 10^−10^ m^2^s^−1^, IAA: 18.72 × 10^−10^ m^2^s^−1^) than IAA (Figure 8d,e), so JXGZ can quickly diffuse from the oil phase to the oil–water interface to replace IAA. It further shows that the affinity between JXGZ and water is stronger than between IAA and water. Moreover, IAA can absorb on the oil–water interface through hydrogen bonds and π-π stacking of aromatic rings in IAA (Figure 8f), and the N and O atoms in JXGZ can form hydrogen bonds with hydrogen in the water molecules at the oil–water interface (Figure 8g). These hydrogen bond energies are further calculated by the all-atom molecular dynamics simulation (Table 2 and Table 3). It was found that the hydrogen bond energies formed between the O and N atoms in the JXGZ and water molecules is higher than that between the IAA molecules, and the hydrogen bond energies between the IAA and water molecules. When the JXGZ was introduced into the emulsions system, the intramolecular hydrogen bonding energies in the IAA molecules and the intermolecular hydrogen bonding energies between the IAA molecules and water molecules are significantly reduced (Table 3). In addition, the intramolecular interaction energy in the IAA molecules before and after the addition of JXGZ were calculated by the Materials Studio software. It shows that the presence of JXGZ could significantly reduce the intramolecular interactions in IAA from 287.651 kJ·mol^−1^ to 40.512 kJ·mol^−1^. Based on the results, it can be inferred that the non-bonding energy of π-π stacking in the IAA molecules and the interaction between fatty chains are significantly reduced at the oil–water interface.

The above results show that JXGZ can form hydrogen bonds with hydrogen in water molecules at the oil-water interface through N and O atoms, thereby replacing IAA molecules at the oil-water interface. As shown in Figure 9, different types of hydrogen bonds include oxygen hydrogen bond and nitrogen hydrogen bond. The oxygen and nitrogen in JXGZ and the hydrogen in water molecules can form new double type hydrogen bonds (defined as the dual swords synergistic effect under hydrogen bond reconstruction). Under the interaction of the dual swords synergistic effect, JXGZ can quickly enter the oil–water interface to destroy the IAA interfacial films, thus quickly realizing the complete demulsification of W/HO emulsions.

## 3. Materials and Methods

### 3.1. Materials

Toluene, xylene, n-heptane, methanol, ethyl ether and p-toluene sulfonic acid were purchased from Chongqing Chuandong Chemical Group Co., Ltd. (Chongqing, China) Benzoyl peroxide (BPO), acrylic acid, ethylenediamine (EDA), methyl acrylate (MA), sulfoxide chloride, pyridine and N, N-dimethylformamide (DMF) were purchased from Aladdin Chemical Reagent Corporation (Shanghai). Fatty alcohol nonionic polyether (FAP) with hydroxyl terminated was purchased from Hai’an Petrochemical Plant. The physical photo of FAP is shown in Appendix A. The milli-Q water (with resistance of 18.0 MΩ.cm at 25 °C) was used in this work. The interfacially active asphaltenes (IAA) were extracted using the procedures given elsewhere [14], and the analysis of the molecular structure characteristics of IAA adopts our previous research results [17].

### 3.2. Synthesis of the Novel Demulsifier (JXGZ)

According to relevant research reports, with MA and EDA as starting materials, the hyperbranched poly(amido amine) (h-PAMAM) was economically synthesized through a Michael addition-polymerization reaction [38]. The physical photo of h-PAMAM was shown in Appendix A. Then, according to our previous research results, oxygen-rich non-ionic polyether (MJTJU-2) was synthesized by esterification and polymerization [31]. The physical photo of MJTJU-2 is shown in Appendix A. The synthetic route of the novel demulsifier simultaneously containing N and O with strong hydrogen bonding (JXGZ) was illustrated in Figure 10. Firstly, MJTJU-2 was added into a three-necked flask. Then, sulfoxide chloride (80 mL) was slowly added into a three-necked flask and DMF (5 mL) was dropped into a three-necked flask. The mixed solution was subjected to acyl chlorination at 70 °C for 24 h. After the reaction, the excess solvent was removed by vacuum distillation. Then, the product was added after the acyl chlorination reaction into DMF, dispersed it by ultrasound at 60 °C for 30 min and poured it into a three-necked flask. Next, the mixed solution of 0.5 mL pyridine, 100 mLDMF and 1 g h-PAMAM were added into the three-necked flask and stirred at room temperature for 24 h to carry out the amidation reaction, and the excess solvent was removed by vacuum distillation. Finally, the product was dried in vacuum for 12 h to obtain the JXGZ demulsifier (Appendix A).

### 3.3. Characterization and Analysis of Demulsifiers

The molecular weights of FAP, h-PAMAM and JXGZ were measured by the GPC 50 gel permeation chromatograph (Agilent company, Santa Clara, CA, USA) with standard polystyrene and chromato-graphically pure tetrahydrofuran as the eluent.

The Fourier Transform Infrared Spectroscopy (FTIR, Bio-Rad, FTS6000, Hercules, CA, USA) was used to analyze the functional groups of the FAP, h-PAMAM and JXGZ at room temperature. Nuclear Magnetic Resonance (NMR) characterization of the demulsifiers was performed on a Bruker AVANCE III 500 MHz NMR spectrometer with deuterated chloroform as the solvent.

The thermal stability of the demulsifiers was analyzed by a thermo gravimetric analyzer (TG 209 F3, NETZSCH, Selb, Germany). Demulsifiers were placed in a TGA furnace with the temperature increasing from 25 to 800 °C under a N_2_ atmosphere at the heating rate of 10 °C/min.

The interfacial activity of the demulsifiers was judged by the surface tension and interfacial tension, which can be obtained by using an interfacial tensiometer (FCA2000A4R) [43]. The solutions with different amounts of FAP, h-PAMAM and JXGZ were used as the water phase. Toluene was used as the oil phase. Using an airtight syringe with a fine needle, an aqueous phase was generated in a quartz tube filled with an oil phase [44].

### 3.4. Demulsification Performance Test

To detect the demulsification performance of JXGZ in demulsifying IAA-stabilized W/HO emulsions, W/HO emulsions were prepared according to the following procedures. Dissolving 1 g of IAA in 100 mL of toluene as the oil phase, and 10 mL of ultrapure water as the water phase. Under high-speed stirring (17,000 rpm) under the homogenizer, add the water phase dropwise to the oil phase and perform emulsification for 10 min. Keep W/HO emulsions stable for 12 h. Then, transfer 15mL W/HO emulsions phase into a stoppered cylinder. The different demulsifiers were added to the emulsions at the certain concentration. The measuring cylinder with a stopper is vibrated up and down 120 times to fully mix the demulsifier and emulsions, and then placed in a water bath for the demulsification test. Emulsions without demulsifier was the blank control group. The demulsification efficiency was evaluated by the dehydration ratio (DR) based on Formula (1) [45]. The demulsification rate (Dr) of the emulsions is obtained by the Formula (2).
(1)$$DR=\frac{V_{s}}{V_{i}}\times 100\%$$
(2)$$Dr=\frac{V_{s}}{t_{s}}$$
where *V_s_* is the volume of the separated water, *V_i_* presents the volume of the initial water (10 mL) and *t_s_* is the settling time (min).

### 3.5. Molecular Dynamics Simulation of Interfacial Interactions and Demulsification

***Construction of an all-atom molecular structure model.*** The molecules included in the all-atom molecular computing system are water, toluene, IAA and JXGZ. According to our previous research, the typical molecular monomer structure of IAA was obtained through characterization and analysis of the molecular structure characteristics [29], based on the polymerization degree of the IAA monomer (1.87) [17]. In order to facilitate the construction of a complete IAA molecular calculation model, we use the rounding principle for data selection and take the degree of polymerization of the IAA monomer as 2. In this paper, a complete molecular structure model of IAA is constructed (the molecular formula of IAA is C_144_H_182_N_2_O_6_S_8_). In addition, based on the previous research results [18,38], and structural characteristics of JXGZ, we constructed the demulsifier containing nitrogen and oxygen as the molecular structure model for the model demulsifier. The optimized molecular structure models are shown in Figure 11. The change curve of energy with the simulation steps of the optimized molecular structures are shown in Appendix A.

***Simulation of Interfacial interactions.*** The Forcite module in the Materials Studio software is used to study the interfacial interactions between IAA and JXGZ at the heavy oil–water interface. The heavy oil–water interface model was constructed as shown in Figure 12a,b, which is calculated by all-atom molecular dynamics. A rectangular simulation box of 3.79 × 3.79 × 9.51 nm^3^ was constructed. The box is filled with 100 water, 400 toluene and 2 IAA molecules. The system without demulsifiers has a total of 5784 atoms, while the system with demulsifiers has a total of 6092 atoms. 

The Forcite module simulation details are as follows: Firstly, optimizing the geometric structure of the heavy oil–water interface system model. Setting calculation accuracy was ultra-fine. Molecular dynamics (MD) simulation was carried out at the NVT-ensemble (constant number of atoms (N), volume (V) and temperature (T)). Temperature and pressure were set at 298.15 K and 101.325 kPa, respectively, using the algorithm described by Berendsen et al. (1984). The force field is set to COMPASS II [46]. Van der Waals and electrostatic were all atom based [47,48]; the cut-off distance is set to 1.85 nm. The dynamic time and time step were set to 500 ps and 1 fs, respectively [49]. The system was equilibrium when the energy and temperature fluctuated around a constant value [50] (Appendix A).

***Simulation of demulsification.*** The dissipative particle dynamics (DPD) simulation technique in the Materials Studio software has been used to study the mesoscopic dynamic demulsification process of emulsions [51]. The important step of using DPD to perform simulation is the coarse-graining of the all-atom molecular structure model [41]. Figure 12c–e are the selected molecular structure models coarse-grained with different beads. Thereinto, one IAA molecule, one toluene molecule, three water molecules and one demulsifier molecule are replaced with 15 beads, 2 beads, 1 bead and 15 beads, respectively [42]. The repulsive force parameters were calculated by the Blends module in the Materials Studio software [41]. The repulsive force parameters were shown in Appendix A.

The DPD simulations were carried out using the Mesocite module in the Materials Studio software, version 2020 [52]. A schematic diagram of the DPD calculation process is shown in Appendix A. The simulations were achieved in a cubic box (Figure 12b); the size of box is 24 × 24 × 24 nm^3^. The total number of beads contained in the box was 4.63 × 10^5^, and the periodic boundary temperature was 333.15 K in the box. Based on our previous research [18], the length scale (Rc), mass scale (m) and time scales (Ʈ) in physical units were 6.46 Å, 54 amu and 8.8 ps, respectively, based on the real experimental data. The composition of each component inside the box was as follows: 1.04 wt% IAA, 10.16 wt% water, 88.74 wt% toluene and 0.06 wt% demulsifier. In order to ensure simulation balance, the DPD simulated time step was set as 10,000 ps. The determination of system simulation equilibrium was shown in Appendix A.

### 3.6. Analysis of Dual Swords Synergistic Effect under Hydrogen Bond Reconstruction

In order to understand the roles of JXGZ in demulsifying the emulsions, the dual swords synergistic effect of JXGZ under hydrogen bond reconstruction is described from three aspects of interaction energy, mean square displacement (MSD) and hydrogen bonds. The interaction energy and hydrogen bonds can be described by the radial distribution function (RDF) and hydrogen bonding energies, respectively. The RDF can be calculated using Formula (3) [53,54,55]. The RDF refers to the distribution probability of other particles in space given the coordinates of a certain particle. In addition, the diffusion coefficient is used to describe the diffusion behavior of different molecules in the system from the oil phase to the oil–water interface. The diffusion coefficient is generally calculated based on the mean square displacement (MSD). The MSD can be calculated using Formula (4), and one-sixth of the slope of the time varying curve of the MSD is the diffusion coefficient [56,57]. Herein, the RDF of the emulsions system and the various demulsifier systems are calculated by the Materials Studio software. The MSD of the different molecules (JXGZ and IAA) in the demulsification system are also calculated by the Materials Studio software.
(3)$$\rho_{ij}\left(r\right)=\frac{\left[{∆N}_{ij}\left(r\to r+∆r\right)\right]V}{4\pi \cdot r^{2}∆rN_{i}N_{j}}$$
(4)$$D=\frac{1}{6}\underset{t\to \infty }{\lim}\frac{d}{dt}\sum_{i=1}^{N}\left[{\left|r_{i}\left(t+t_{0}\right)-r_{i}\left(t_{0}\right)\right|}^{2}\right]$$
where *ρ*_ij_ (*r*) is RDF, *V* is the volume of the calculation system, *N_i_* and *N_j_* are the number of different particles, respectively, and Δ*N_ij_*(*r* → *r* + Δ*r*) is the ensemble averaged number of *i* around *j* within a shell from *r* to *r* + Δ*r*. D is MSD; *r_i_*(*t*_0_) and *r_i_* (*t* + *t*_0_) are the vector positions of the particles at *t*_0_ and *t* + *t*_0_, respectively.

The hydrogen bonding energies is calculated as follows [58,59]: Firstly, the Forcite module is used to conduct the atomic molecular dynamics calculation. Then, hydrogen bonds are detected through the hydrogen bond detection module of the Materials Studio software. Finally, hydrogen bonding energies were calculated through energy dynamics calculation.

## 4. Conclusions

A novel demulsifier (JXGZ) with strong hydrogen bonding interaction has been successfully synthesized through esterification, polymerization and amidation reactions. The JXGZ demulsifier possesses high interfacial activity with the average molecular weight of 6504 g/mol. Thereinto, the relative molecular weight distribution (Mw/Mn) of JXGZ (1.16) is lower than that of FAP (1.39) and h-PAMAM (1.31). Our research results indicate that JXGZ possesses an excellent demulsification performance in IAA-stabilized W/HO emulsions. This demulsifier is proven to be effective in quickly breaking the IAA-stabilized emulsions. In total, 100% of water could be dehydrated from the IAA-stabilized emulsions in only 4 min by this demulsifier at its concentration of 400 ppm and 55 °C. The excellent performance of JXGZ in demulsifying the W/HO emulsions is mainly attributed to the abundantly polar hydrophilic groups containing oxygen and nitrogen (i.e., hydroxyl group, amine groups, ester group, carboxyl group and ether bond) in the JXGZ molecules. Based on the molecular dynamics analysis, during the demulsification process, compared with those formed by heteroatoms (O, N, S) in IAA molecules and hydrogen in water molecules, stronger hydrogen bonds can be formed between oxygen and nitrogen containing groups in the JXGZ molecules and water molecules. The JXGZ also performs well in reducing the interactions between the IAA and water molecules, breaking some of the π-π stacking and the intramolecular hydrogen bonds. In a word, the oxygen and nitrogen in JXGZ and the hydrogen in water molecules can form new stronger double-type hydrogen bonds (dual swords synergistic effect under hydrogen bond reconstruction). Under the dual swords synergistic effect, JXGZ can quickly adsorb on the surface of water molecules and further destroy IAA films, thus promoting the aggregation and coalescence of water droplets, and finally accomplishing the demulsification of emulsions. These findings indicate that the JXGZ demulsifier shows great engineering application prospects in the demulsification of heavy oil–water emulsions, and this work provides the key information for developing more efficient chemical demulsifiers suitable for large-scale industrial applications.

## Figures and Tables

**Figure 1 ijms-24-14805-f001:**
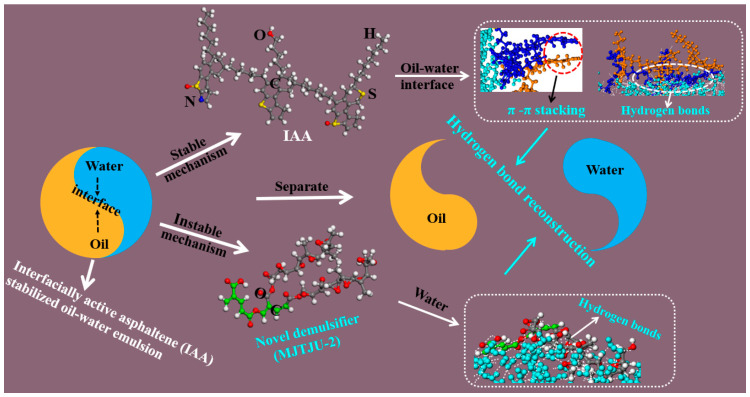
The hydrogen bond reconstruction mechanism of MJTJU-2 demulsifier in IAA-stabilized oil–water emulsions. Reproduced from Ref. [18] with permission of the Elsevier.

**Figure 2 ijms-24-14805-f002:**
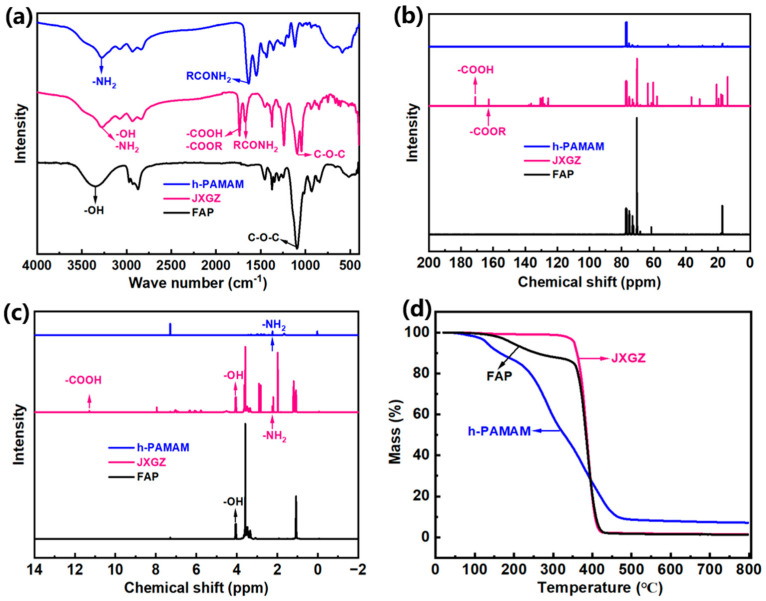
(**a**) FTIR spectra of different demulsifiers, (**b**) ^13^C NMR spectra of different demulsifiers, (**c**) ^1^H NMR spectra of different demulsifiers and (**d**) thermogravimetric analysis spectra of FAP, h-PAMAM and JXGZ.

**Figure 3 ijms-24-14805-f003:**
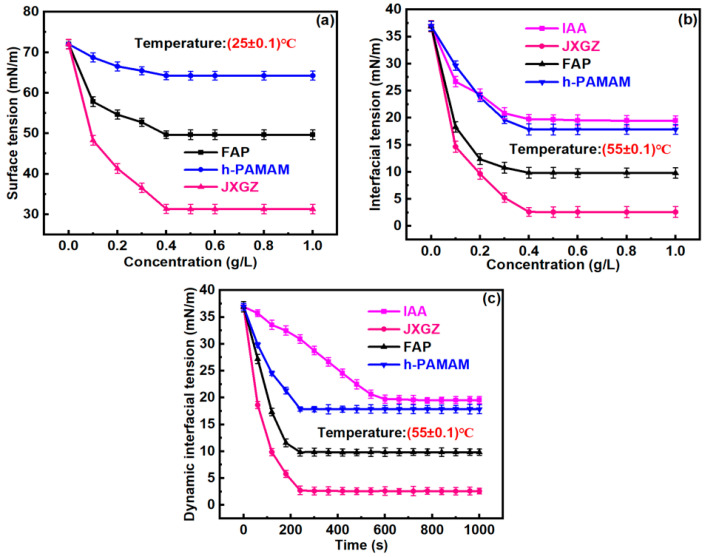
Analysis of interfacial activity of different demulsifiers ((**a**): the changes of surface tension; (**b**): the changes of interfacial tension; (**c**): the changes of dynamic oil–water interfacial tension).

**Figure 4 ijms-24-14805-f004:**
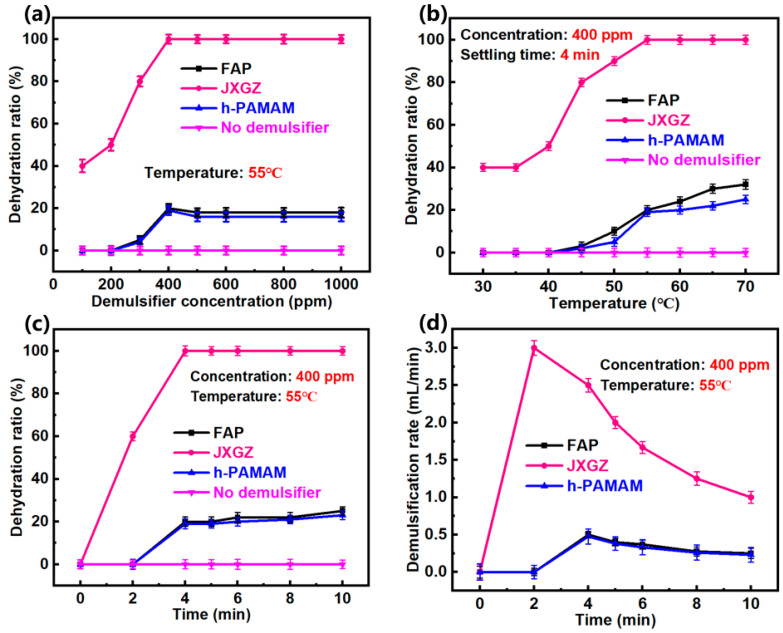
Demulsification performance of demulsifiers in W/HO emulsions under different conditions (**a**): demulsifier concentration; (**b**): demulsification temperature; (**c**): demulsification time; (**d**): dehy-dration rate).

**Figure 5 ijms-24-14805-f005:**
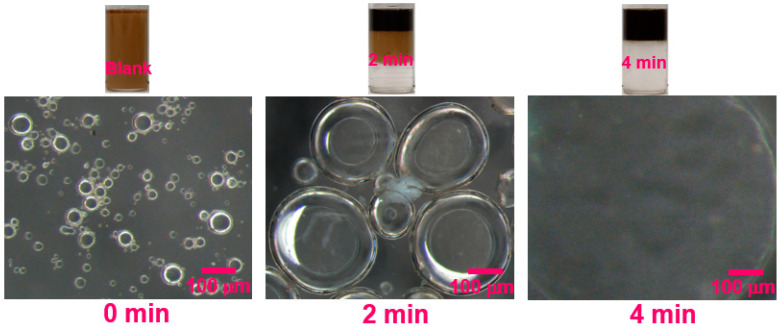
Water separation photographs and the stereomicroscopic photos of W/HO emulsions at 55 °C by using JXGZ demulsifier.

**Figure 6 ijms-24-14805-f006:**
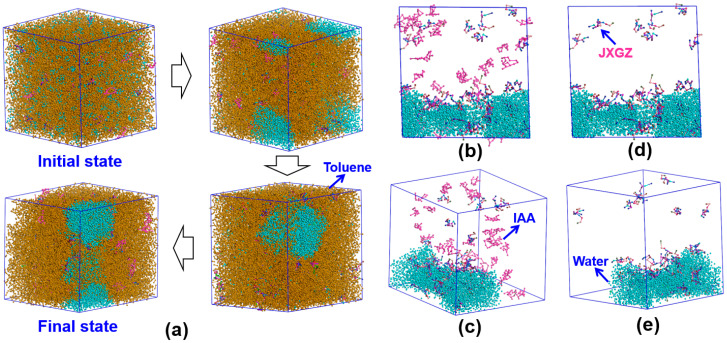
Dynamic demulsification process of W/HO emulsions ((**a**): mesoscopic dynamic demulsification process; (**b**–**e**): display demulsifiers, IAA and water molecules).

**Figure 7 ijms-24-14805-f007:**
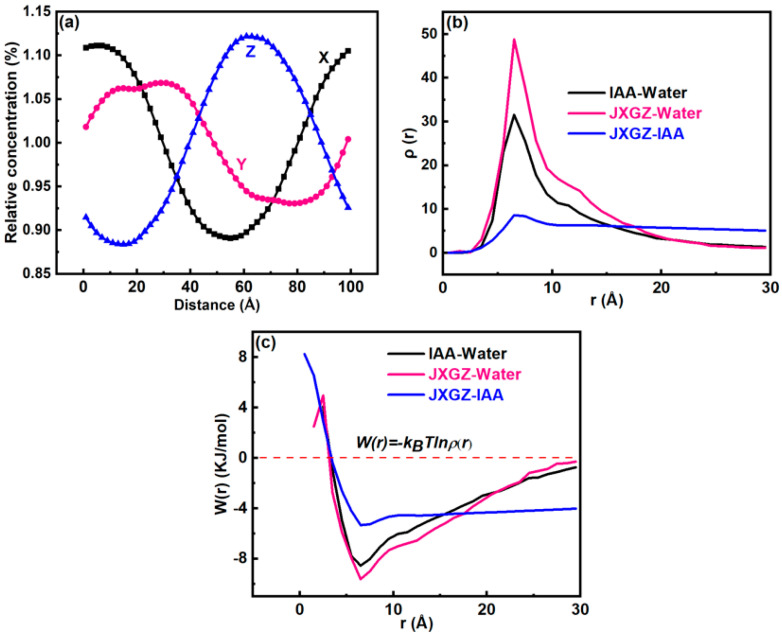
(**a**) The variation of water concentration during dynamic demulsification process in X, Y and Z directions; (**b**) the RDF (adial distribution function); (**c**) the PMF (potential of mean force).

**Figure 8 ijms-24-14805-f008:**
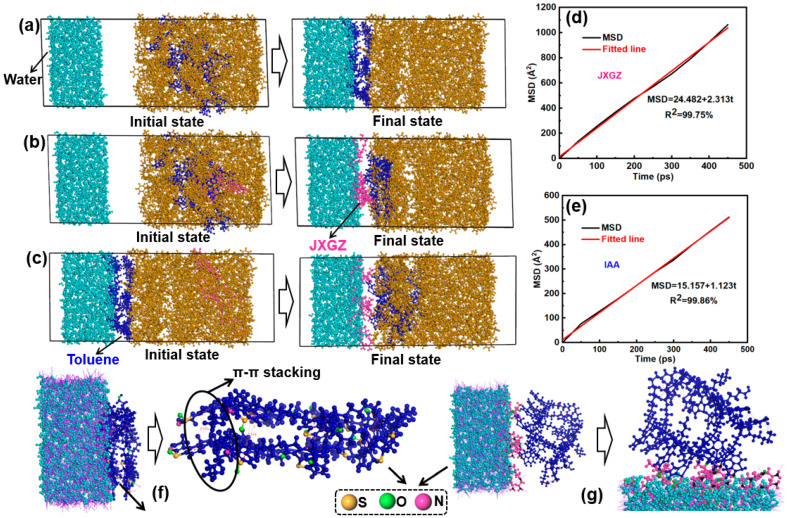
(**a**–**c**) Adsorption state of IAA and JXGZ at the oil–water interface; (**d**,**e**) mean square displacement; (**f**,**g**) hydrogen bonds at the oil water interface.

**Figure 9 ijms-24-14805-f009:**
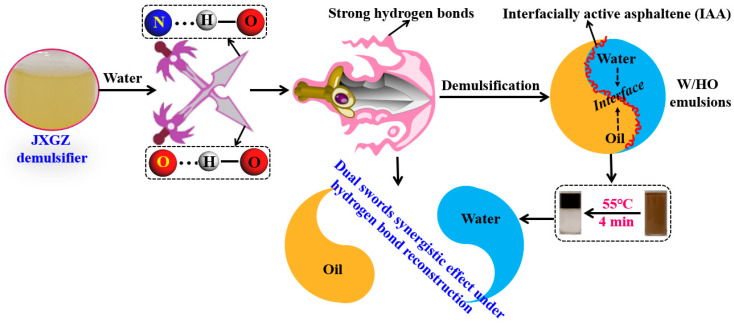
Schematic diagram of the synergistic effect mechanism of the dual swords synergistic effect.

**Figure 10 ijms-24-14805-f010:**
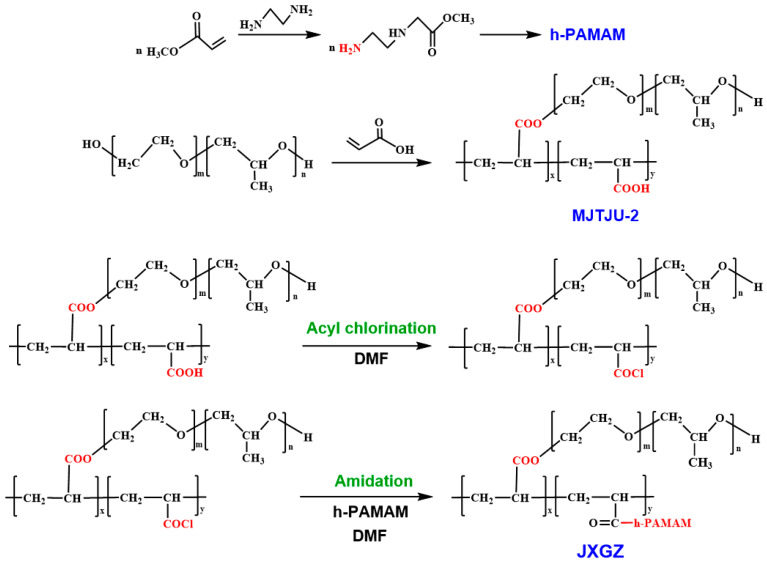
The synthetic route of JXGZ.

**Figure 11 ijms-24-14805-f011:**
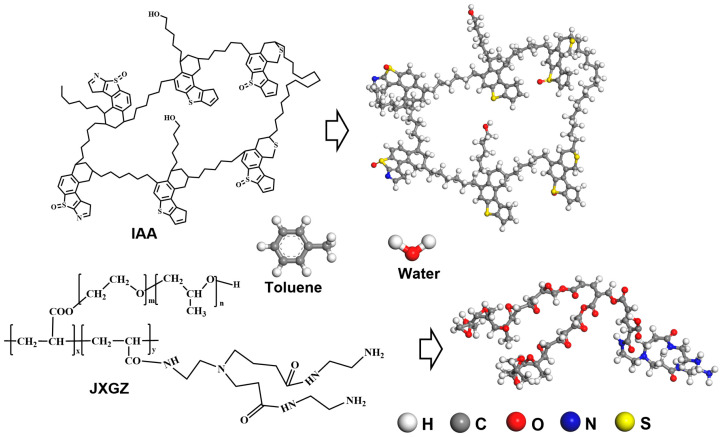
The optimization models of chosen molecular structures.

**Figure 12 ijms-24-14805-f012:**
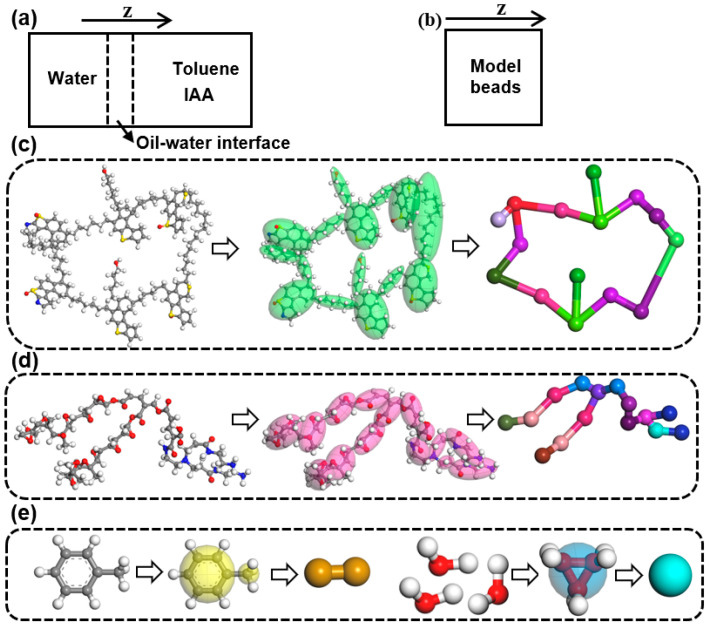
Schematic diagram of calculation model system (**a**,**b**), and all-atom molecular structure models and mesoscopic molecular structure models ((**c**): IAA, (**d**): JXGZ, (**e**): toluene and water).

**Table 1 ijms-24-14805-t001:** The GPC analysis results of demulsifiers.

Demulsifiers	M_w_ (g/mol)	M_n_ (g/mol)	Mw/Mn
FAP	3754	5215	1.39
JXGZ	6504	5620	1.16
h-PAMAM	10896	8296	1.31

**Table 2 ijms-24-14805-t002:** The hydrogen bonding energies within IAA molecules and between IAA and water molecules.

System	Functional Group	Hydrogen Bond	Bond Energy (kJ·mol^−1^)
IAA molecules	S=O	O–H…Y	48.3
	–OH	O–H…Y	39.8
	N–R	N–H…Y	26.6
	S=O	S–H…Y	20.2
	S–R	S–H…Y	16.6
IAA and water	S=O	O…H–O	37.2
	–OH	O…H–O	21.6
	N–R	N…H–O	18.6
	S=O	S…H–O	11.4
	S–R	S…H–O	6.1

**Table 3 ijms-24-14805-t003:** The hydrogen bonding energies between JXGZ and water molecules, the hydrogen bonding energies within IAA molecules and the hydrogen bonding energies between IAA and water molecules after adding JXGZ demulsifiers.

System	Functional Group	Hydrogen Bond	Bond Energy (kJ·mol^−1^)
JXGZ and water	COOH (Double bond)		60.6
	COOH (Single bond)		56.5
	COOR (Double bond)		56.3
	COOR (Single bond)	O…H–O	55.4
	C–O–C (PEO)		50.6
	–OH		51.5
	C–O–C (PPO)		16.3
	–CONH–	N–H…Y	52.4
	–NH_2_	N–H…Y	58.7
	–CONH–	N–H…Y	54.6
	–CNC–	N–H…Y	49.5
IAA intermolecular	S=O	O–H…Y	19.3
	–OH	O–H…Y	16.5
	N–R	N–H…Y	11.3
	S=O	S–H…Y	9.9
	S–R	S–H…Y	6.5
IAA and water	S=O	O…H–O	9.3
	–OH	O…H–O	6.8
	N–R	N…H–O	2.3
	S=O	S…H–O	1.0
	S–R	S…H–O	0.9

Notes: PEO is polyethylene oxide. PPO is polypropylene oxide.

## Data Availability

Not Applicable.

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
