# Peer review of "A Novel Demulsifier with Strong Hydrogen Bonding for Effective Breaking of Water-in-Heavy Oil Emulsions"

_ijms, 2023, doi:10.3390/ijms241914805_

Round 1

Reviewer 1 Report

Manuscript ID: ijms-2594839

Title: A novel demulsifier with strong hydrogen bonding for effective breaking of water-in-heavy oil emulsions

Authors: Xiao Xia, Jun Ma *, Fei Liu, Haifeng Cong, Xingang Li

Comments:

The manuscript reports the efficiency of demulsification of a new demulsifier (JXGZ) obtained by functionalization of a previously synthesized demulsifier (MJTJU-2) with active NH and NH2 groups. The effectiveness of JXGZ has been experimentally proven. Using MD simulation, the mechanisms responsible for the interaction of JXGZ with water and a model interfacially active asphaltene were simulated.

The manuscript is easy to read. The topic of the study is relevant in the field and its motivation is justified. However, I have some critical remarks to the manuscript.

JXGZ does not contain hydroxyl groups. Their presence indicates that the functionalization of MJTJU-2 was not complete. Therefore, JXGZ is not “a novel nitrogen and oxygen containing demulsifier. It is a demulsifier containing amino and amido groups.

What meaning the authors put into the term "dual swords synergistic effect" is not clear.

Please explain in detail how the specific hydrogen bond energies presented in Tables 2 and 3 were estimated.

Please report if JXGZ remains in the water or oil phase.

Conventionally, the chemical shift scale is from right (negative values or zero) to left (positive values), see Figure 5.

“Based on the polymerization degree (1.87) of IAA monomer is about 2.” The meaning of the sentence is not clear.

Author Response

Referee’s comments:

Reviewer: #1

Comments:

The manuscript reports the efficiency of demulsification of a new demulsifier (JXGZ) obtained by functionalization of a previously synthesized demulsifier (MJTJU-2) with active NH and NH2 groups. The effectiveness of JXGZ has been experimentally proven. Using MD simulation, the mechanisms responsible for the interaction of JXGZ with water and a model interfacially active asphaltene were simulated. The manuscript is easy to read. The topic of the study is relevant in the field and its motivation is justified. However, I have some critical remarks to the manuscript.

Thanks the reviewer very much for your evaluation and recognition to our work. We accept this excellent suggestion. We simplify the relevant presentation. Some revisions have been made in the revised manuscript (MS), also shown as below. We believe the revised manuscript would be much more readable and accurate.

1) JXGZ does not contain hydroxyl groups. Their presence indicates that the functionalization of MJTJU-2 was not complete. Therefore, JXGZ is not “a novel nitrogen and oxygen containing demulsifier. It is a demulsifier containing amino and amido groups.

Many thanks for the reviewer’s comment. At present, the demulsifier used for demulsification of heavy oil-water emulsions in the petroleum industry is mainly non-ionic polyether. It is a polymer terminated with hydroxyl groups, and it possesses significant demulsification effect on demulsification of heavy oil-water emulsions. In this paper, MJTJU-2 and JXGZ are also polymers with a certain molecular weight distribution, and the reaction of the polymer cannot be completely completed. Only some functional groups will react. In addition, nuclear magnetic resonance (NMR) characterization also indicates that the polymer is not a pure substance. Therefore, MJTJU-2 and JXGZ both contain hydroxyl groups.

2) What meaning the authors put into the term "dual swords synergistic effect" is not clear.

We thank the reviewer for this good question. JXGZ molecules could replace the IAA molecules adsorbed on the surface of water molecules mainly by different types of hydrogen bonds. As shown in Fig. R1, different types of hydrogen bonds include oxygen hydrogen bond and nitrogen hydrogen bond. The oxygen and nitrogen in JXGZ and the hydrogen in water molecules can form new stronger double type hydrogen bonds (defined as dual swords merge effect under hydrogen bond reconstruction). Under the interaction of dual swords synergistic effect, JXGZ can quickly adsorb on the surface of water molecules and further destroy IAA films, thus promoting the aggregation and coalescence of water droplets, and finally accomplishing the demulsification of IAA-stabilized W/HO emulsions. We have added corresponding mechanism diagrams in the manuscript. (Section 3.4: pages 15 and 16 in the revised manuscript)

Fig. R1. Schematic diagram of the synergistic effect mechanism of dual swords synergistic effect.

3) Please explain in detail how the specific hydrogen bond energies presented in Tables 2 and 3 were estimated.

We thank the reviewer for this good question. The hydrogen bond energy is calculated as follows: first, the simulation system was calculated by all atom molecular dynamics, then hydrogen bonds was detected by the hydrogen bond detection module of material studio software, finally the hydrogen bonds energy were calculated by energy dynamics calculation. We have already provided relevant explanations in section 2.6 of the manuscript.

4) Please report if JXGZ remains in the water or oil phase.

Thanks the reviewer again for bringing this good question. Our previous research has shown that MJTJU-2 is an amphiphilic (hydrophilic and lipophilic) surfactant that can be dissolved in water or oil (toluene)[1]. On the basis of MJTJU-2, introducing N-containing groups into MJTJU-2 to obtain JXGZ, thereby enhancing its hydrogen bonding effect and improving its demulsification performance for demulsification of water-in-heavy oil. In addition, as shown in Fig. 9 and 10 in the manuscript, the calculation results of hydrogen bond energy, the average force potential between JXGZ and water molecules, and the kinetic calculation of demulsification process indicate that JXGZ possesses stronger interaction with water molecules. Therefore, JXGZ is also an amphiphilic demulsifier, and after its demulsification is completed, most JXGZ enters into the water phase, while a few enters into the oil phase.

5) Conventionally, the chemical shift scale is from right (negative values or zero) to left (positive values), see Figure 5.

Good comments from the reviewer. We have revised Fig. 5 as shown in Figure R2. (Page 9 in revised manuscript)

Figure R2. (a) FTIR spectra of demulsifier samples, (b)13C NMR spectra of demulsifier samples, (c)1H NMR spectra of demulsifier samples, (d) Thermogravimetric analysis spectra of FAP, h-PAMAM and JXGZ demulsifiers.

6)“Based on the polymerization degree (1.87) of IAA monomer is about 2.” The meaning of the sentence is not clear.

Thanks the reviewer again for bringing this good question. Our previous research has shown that the  degree of polymerization of IAA monomer is 1.87[2]. In order to facilitate the construction of a complete IAA molecular calculation model, we use the rounding principle for data selection and take the degree of polymerization of IAA monomer as 2. We have revised the relevant expressions in the manuscript. (Page 5 in revised manuscript)

References

[1]        Ma J, Li X, Zhang X, Sui H, He L, Wang S. A novel oxygen-containing demulsifier for efficient breaking of water-in-oil emulsions. Chemical Engineering Journal 2020; 385: 123826.

[2]        Li X, Ma J, Bian R, Cheng J, Sui H, He L. Novel polyether for efficient demulsification of interfacially active asphaltene-stabilized water-in-oil emulsions. Energy & Fuels 2020; 34: 3591-3600.

Reviewer 2 Report

This is an interesting study that deserves to be published.

However several corrections are needed to improve this manuscript before the full acceptance.

The list of reservations from my part is given below.

- Eqs. 3 and 4 are not explained in detail, for example the explanation of meanings of Ni and Nj is not suffiecient;

- the H-bond energy is calculated, I think that referring to references is not enough, it should be shortly described in this manuscript how these energies are calculated;

- Table 2, for bond energies the three decimals are given, one is enough since it better corresponds to the accuracy of calculations;

- numerous figures that appear in this study are not described sufficiently, I suggest move part of them to Supplementary Material and to describe remaining ones in detail.

Author Response

Reviewer: #2

Comments:

This is an interesting study that deserves to be published.

However several corrections are needed to improve this manuscript before the full acceptance.

The list of reservations from my part is given below.

1) - Eqs. 3 and 4 are not explained in detail, for example the explanation of meanings of Ni and Nj is not suffiecient;

Many thanks for the reviewer’s comment. The radial distribution function (RDF) refers to the distribution probability of other particles in space given the coordinates of a certain particle. Equation 3 is the defining formula for solving the RDF. mean square displacement (MSD) describes the changes in the motion position of particles in the system. Equation 4 is the defining formula for solving the MSD. Both equation 3 and equation 4 have been reported. In addition, the RDF and MSD can be calculated by Materials Studio software. We have revised section 2.6 in the revised manuscript. As shown below:

2.6. Analysis of dual swords synergistic effect under hydrogen bond reconstruction

To understand the roles of JXGZ in demulsifying the IAA-W/O emulsions, the dual swords synergistic effect of JXGZ under hydrogen bond reconstruction is described from three aspects of interaction energy, mean square displacement (MSD) and hydrogen bonds. The interaction energy and hydrogen bonds function can be described by radial distribution function (RDF) and hydrogen bond energy, respectively. The RDF can be calculated using formula (3) [1-3]. The RDF refers to the distribution probability of other particles in space given the coordinates of a certain particle. In addition, the diffusion coefficient is used to describe the diffusion behavior of different molecules in the system from the oil phase to the oil-water interface. The diffusion coefficient is generally calculated based on mean square displacement (MSD). The MSD can be calculated using formula (4), and one sixth of the slope of the time varying curve of the MSD is the diffusion coefficient[45]. Herein, The RDF of the W/O emulsions system and the different demulsifier systems are calculated by Materials Studio software. Besides, the MSD of the different molecules (JXGZ and IAA) in demulsification system are also calculated by Materials Studio software.

              (3)

(4)

Where rij (r) is RDF, V is the system volume, Ni and Nj are number of different particles, respectively, and ΔNij(r→r + Δr) is the ensemble averaged number of i around j within a shell from r to r + Δr. D is MSD, ri(t0) and ri (t+t0) are the vector positions of the particles at t0 and t+t0, respectively.

The hydrogen bond energy is calculated as follows [67]: first, the simulation system was calculated by all atom molecular dynamics, then hydrogen bonds was detected by the hydrogen bond detection module of material studio software, finally the hydrogen bonds energy were calculated by energy dynamics calculation.

2)- the H-bond energy is calculated, I think that referring to references is not enough, it should be shortly described in this manuscript how these energies are calculated;

We thank the reviewer for this good question. The hydrogen bond energy is calculated as follows: first, the simulation system was calculated by all atom molecular dynamics, then hydrogen bonds was detected by the hydrogen bond detection module of material studio software, finally the hydrogen bonds energy were calculated by energy dynamics calculation. We have already provided relevant explanations in section 2.6 of the manuscript.

3)- Table 2, for bond energies the three decimals are given, one is enough since it better corresponds to the accuracy of calculations;

Thanks the reviewer for this excellent suggestion. As suggested, we have reorganized the table data in the revised manuscript according to scientific data selection rules.

4) - numerous figures that appear in this study are not described sufficiently, I suggest move part of them to Supplementary Material and to describe remaining ones in detail.

Thanks the reviewer for pointing out this good question. We would like to reiterate that this article is a further extension of our previous research[1,2]. To ensure the logic of the entire article, the figures in the text are currently essential. In addition, we have included some supplementary explanatory figures in the supplementary materials.

References

[1]        Ma J, Li X, Zhang X, Sui H, He L, Wang S. A novel oxygen-containing demulsifier for efficient breaking of water-in-oil emulsions. Chemical Engineering Journal 2020; 385: 123826.

[2]    Ma J, Yang Y, Li X, Sui H, He L, Mechanisms on the stability and instability of water-in-oil emulsion stabilized by interfacially active asphaltenes: Role of hydrogen bonding reconstructing, Fuel 2021; 297: 120763.

Reviewer 3 Report

This paper describes an experimental study of breaking up heavy oil-water emulsions through the development of a new demulsifier. They then use classical MD simulations to explain the results.

Please do not exaggerate. Remove ‘great’ from the abstract in terms of the application.

The paper needs quite a bit of work on the English. See figure 3 caption as an example and the entire computational methods section.

The entire computational section has very poor referencing. Explain what is in the forcite module as this is not known to the readers. Material Studio software needs to be capitalized and references given to it and to all of the approaches in all of the modules. For example, provide a reference to the COMPASS II force field. What do they mean by group-based van der Waals and electrostatic cutoffs?

How many atoms? How was the system equilibrated. 500 ps is too short for such a simulation. Run at least 1 nsec.

Provide references to DPD and 3explain better.

The RDF does not give any energetic results.

I do not think that the reference numbering is correct. Ref. 48 and 49 do not have anything to do with hydrogen bond energies. How were the hydrogen bonds detected? What criteria?

How were the IR assignments made? Provide references. The OH band at 3400 cm-1 seems to be too low to me for such a band not involved in hydrogen bonding to water. The peak at 3284 cm-1 is not likely to contain an OH group unless it is strongly hydrogen bonded.

The NMR assignment section needs referencing as well.

Please provide error bars where possible.

Under figure 5. I doubt they have percentages to two decimal place accuracy. On the same page, they do not have interfacial tensions up to 5 significant figures. This all needs to be addressed.

Figures 6 and 7 need error bars.

In the description of figure 9 on p. 12, the interaction energies are not good to even 1 kj/mol much less to two decimal places in kJ/mol.

Figure 9 is too small.

Figure 10 should have a cutoff at no more than 30 angstroms to better show what is happening in the RDF.

On p. 14 and 15 in the tables. The molecular interactions are not good to even 1 kJ/mol. This needs to be fixed.

Just above figure 11. Are these positive or negative interaction energies? How are they defining interaction energies? If it is binding, then they are negative.

The hydrogen bond energies in tables 2 and 3 should be negative or defined as such and only given to one decimal place at best.

The values in table 4 are already in the text and a table is not needed.

What is the distribution about the average molecular weight in the conclusions?

They need to show the hydrogen bonds that they think are present, especially a double type hydiorgen bond. What do they mean by this?

What is the dual swords effect? I never could figure this term out. It is not clear what they mean.

They do not make good connections between the experiment and the calculated results.

This paper needs significant revision.

there are issues with the English.

Author Response

Reviewer: #3

Comments:

This paper describes an experimental study of breaking up heavy oil-water emulsions through the development of a new demulsifier. They then use classical MD simulations to explain the results.

1Please do not exaggerate. Remove ‘great’ from the abstract in terms of the application.

We thank the reviewer for this good suggestion. Done as suggested. (Page 1 in revised manuscript)

2The paper needs quite a bit of work on the English. See figure 3 caption as an example and the entire computational methods section.

Thanks the reviewer for this excellent suggestion again. We have tried our best to polish the language again by removing some redundant content. We also ask for a native speaker to help us to improve the writing. We believe the revised manuscript (MS) would be much more readable and accurate.

3The entire computational section has very poor referencing. Explain what is in the forcite module as this is not known to the readers. Material Studio software needs to be capitalized and references given to it and to all of the approaches in all of the modules. For example, provide a reference to the COMPASS II force field. What do they mean by group-based van der Waals and electrostatic cutoffs?

Good questions from the reviewer. van der Waals, electrostatic and cut off distance are  parameter setting option in forcite module. We have revised the calculation section in revised manuscript. (Pages 5-7 in revised manuscript)

4How many atoms? How was the system equilibrated. 500 ps is too short for such a simulation. Run at least 1 nsec.

Thanks the reviewer for this good question again. The system without demulsifiers has a total of 5784 atoms, while the system with demulsifiers has a total of 6092 atoms. According to relevant research report[4]. The energy and temperature fluctuations of the all-atom molecular dynamics calculation system range from 0.1% to 5%, indicating that the calculation system has reached equilibrium. We have added relevant reference in the revised manuscript.

5Provide references to DPD and 3 explain better.

We thank the reviewer for this good suggestion. Done as suggested.

6The RDF does not give any energetic results.

Thanks the reviewer. RDF indeed cannot obtain energy, however, we calculate the interaction energy based on the potential of mean force.

7I do not think that the reference numbering is correct. Ref. 48 and 49 do not have anything to do with hydrogen bond energies. How were the hydrogen bonds detected? What criteria?

Good comments from the reviewer. We are sorry for the typos. The references we cited are indeed inappropriate, and we have made modifications. The hydrogen bond energy is calculated as follows: first, the simulation system was calculated by all atom molecular dynamics, then hydrogen bonds was detected by the hydrogen bond detection module of Materials Studio software, finally the hydrogen bonds energy were calculated by energy dynamics calculation.

8How were the IR assignments made? Provide references. The OH band at 3400 cm-1 seems to be too low to me for such a band not involved in hydrogen bonding to water. The peak at 3284 cm-1 is not likely to contain an OH group unless it is strongly hydrogen bonded.

Good comments from the reviewer. In terms of infrared spectrum analysis, we have re added relevant references in the manuscript. (pages 7 and 8 in the revised manuscript)

9The NMR assignment section needs referencing as well.

Thanks the reviewer again. Done as suggested.

10Please provide error bars where possible.

Good comments from the reviewer. Done as suggested.

11Under figure 5. I doubt they have percentages to two decimal place accuracy. On the same page, they do not have interfacial tensions up to 5 significant figures. This all needs to be addressed.

We thank the reviewer for this good question. The interfacial tension is five significant digits, and we repeated three measurements to take the average value, and then added an error bars through calculation. We have made corresponding modifications in the revised manuscript. As shown in Figure R1.

Figure R1. Analysis of interfacial activity of different demulsifiers (a: Surface tension of FAP, h-PAMAM and JXGZ demulsifier aqueous solutions; b: interfacial tension of the oil-water interface with FAP, h-PAMAM, JXGZ demulsifiers and IAA; c: dynamic oil-water interfacial tension with FAP, h-PAMAM, JXGZ and IAA).

12Figures 6 and 7 need error bars.

Good comments from the reviewer. Done as suggested.

13In the description of figure 9 on p. 12, the interaction energies are not good to even 1 kj/mol much less to two decimal places in kJ/mol.

Good comments from the reviewer. We have made corresponding modifications in the revised manuscript.

14Figure 9 is too small.

Thanks the reviewer for this excellent suggestion. We have resized Figure 9.

15Figure 10 should have a cutoff at no more than 30 angstroms to better show what is happening in the RDF.

Thanks the reviewer point this question. The interval for analyzing RDF is 50 angstroms. In order to maintain consistency in the initial setting of calculation parameters, we cannot modify it to be less than 30 angstroms, and this does not affect the display effect of the data.

16) On p. 14 and 15 in the tables. The molecular interactions are not good to even 1 kJ/mol. This needs to be fixed.

Thanks the reviewer for this excellent suggestion. As suggested, we have reorganized the table data in the revised manuscript according to scientific data selection rules.

17) Just above figure 11. Are these positive or negative interaction energies? How are they defining interaction energies? If it is binding, then they are negative.

Thanks the reviewer for this good question. The values calculated for interaction energy can be both positive and negative. Since JXGZ and IAA both contain hydrophilic groups, they have a certain attraction to water molecules. The negative sign in this article represents the attraction between molecules.

18The hydrogen bond energies in tables 2 and 3 should be negative or defined as such and only given to one decimal place at best.

Thanks the reviewer. We measure the level of hydrogen bonds by the numerical value of hydrogen bond energies. The calculation of the hydrogen bond energies in this paper is a positive value, and negative signs cannot be added arbitrarily.

19The values in table 4 are already in the text and a table is not needed.

We thank the reviewer for this good suggestion. We have deleted Table 4 and revised the relevant content of the manuscript. (pages 14 and 15 in the revised manuscript)

20What is the distribution about the average molecular weight in the conclusions?

Thanks the reviewer again. According to the theoretical foundation of polymer chemistry, the molecular weight of a polymer is usually described by the average molecular weight (Mw) and number average molecular weight (Mn). The molecular weight distribution is the ratio of the Mw to the Mn. The smaller the ratio, the better the quality of the polymer.

21They need to show the hydrogen bonds that they think are present, especially a double type hydiorgen bond. What do they mean by this?

We thank the reviewer for this good question. This article emphasizes two types of hydrogen bonds, namely the hydrogen bonds formed by oxygen in JXGZ demulsifier and hydrogen in water molecules, and the hydrogen bonds formed by nitrogen in JXGZ demulsifier and hydrogen in water molecules. For a better understanding, we have also added relevant explanations in the mechanism discussion section.

22What is the dual swords effect? I never could figure this term out. It is not clear what they mean.

We thank the reviewer for this good question. JXGZ molecules could replace the IAA molecules adsorbed on the surface of water molecules mainly by different types of hydrogen bonds. As shown in Fig. R1, different types of hydrogen bonds include oxygen hydrogen bond and nitrogen hydrogen bond. The oxygen and nitrogen in JXGZ and the hydrogen in water molecules can form new stronger double type hydrogen bonds (defined as dual swords merge effect under hydrogen bond reconstruction). Under the interaction of dual swords synergistic effect, JXGZ can quickly adsorb on the surface of water molecules and further destroy IAA films, thus promoting the aggregation and coalescence of water droplets, and finally accomplishing the demulsification of IAA-stabilized W/HO emulsions. We have added corresponding mechanism diagrams in the manuscript. (Section 3.4: pages 15 and 16 in the revised manuscript)

Fig. R1. Schematic diagram of the synergistic effect mechanism of dual swords synergistic effect.

23They do not make good connections between the experiment and the calculated results.

Thanks the reviewer. We proved that JXGZ demulsifier possesses efficient demulsification performance for water-in-heavy oil emulsions through experiments. And we verified the interface process of JXGZ replacing IAA through all atom molecular dynamics simulation, and verified the dynamic demulsification process of JXGZ in emulsions through dissipative particle dynamics (DPD) simulation. Therefore, there is a good connection between the experimental part and the theoretical calculation part.

24This paper needs significant revision.

Thanks the reviewer for pointing out this good question. We have made major modifications to the entire paper. We believe the revised manuscript (MS) would be much more readable and accurate.

25there are issues with the English.

Good questions from the reviewer. We have tried our best to polish the language again by removing some redundant content. We also ask for a native speaker to help us to improve the writing.

References

[1]        Ma J, Li X, Zhang X, Sui H, He L, Wang S. A novel oxygen-containing demulsifier for efficient breaking of water-in-oil emulsions. Chemical Engineering Journal 2020; 385: 123826.

[2]        Li X, Ma J, Bian R, Cheng J, Sui H, He L. Novel polyether for efficient demulsification of interfacially active asphaltene-stabilized water-in-oil emulsions. Energy & Fuels 2020; 34: 3591-3600.

[3]    Ma J, Yang Y, Li X, Sui H, He L, Mechanisms on the stability and instability of water-in-oil emulsion stabilized by interfacially active asphaltenes: Role of hydrogen bonding reconstructing, Fuel 2021; 297: 120763.

[4]   Huang H,  Alexander A. Dissipative particle dynamics for directed self-assembly of block copolymers. The Journal of Chemical Physics 2019; 151(15) :  154905.

Round 2

Reviewer 1 Report

I have no more comments that would necessitate another review cycle.

Author Response

Reviewer: #1

Comments:

I have no more comments that would necessitate another review cycl

We thank the reviewer for your thought-provoking comments and suggestions.

Reviewer 3 Report

There are still issues. The authors need to learn to put the answers to the questions in the manuscript, not just in the response.

How many atoms? This was not added to the manuscript.

How was the system equilibrated. This was not added to the manuscript.

500 ps is too short for such a simulation. Run at least 1 nsec. This was not doen. This is a small system. It does not matter what they have published before. It matters what they did in the current work.

Figure 10 should have a cutoff at no more than 30 angstroms to better show what is happening in the RDF. This was not done. I ma not asking them to do a calculation with a 30 angstroms cutoff but to replot the figure. This is easy to do and must be done.

What is the distribution about the average molecular weight in the conclusions? This was not added that I could find. There is a distribution of molecular weights for any polymer, and they need to add this in.

They do not make good connections between the experiment and the calculated results. They need to make a stronger case in the text for this. They did not.

This paper still needs revision.

Improved

Author Response

Comments:

There are still issues. The authors need to learn to put the answers to the questions in the manuscript, not just in the response.

1How many atoms? This was not added to the manuscript.

Thanks the reviewer for this good question. The system without demulsifiers has a total of 5784 atoms, while the system with demulsifiers has a total of 6092 atoms. The relevant contents have been added to section 3.5.

2How was the system equilibrated. This was not added to the manuscript.

Thanks the reviewer. We have added relevant contents in section 3.5 of the revised manuscript.

3500 ps is too short for such a simulation. Run at least 1 nsec. This was not doen. This is a small system. It does not matter what they have published before. It matters what they did in the current work.

Thanks the reviewer for pointing out this good question. For the simulation of all-atom molecular dynamics based on Newtonian mechanics, 500 ps is not small anymore. In addition, 500 ps has reached the simulation equilibrium, which can explain the problem. There is no need to spend more machine time and increase additional costs.

4Figure 10 should have a cutoff at no more than 30 angstroms to better show what is happening in the RDF. This was not done. I ma not asking them to do a calculation with a 30 angstroms cutoff but to replot the figure. This is easy to do and must be done.

Thanks the reviewer for this excellent suggestion. Done as suggested.

5What is the distribution about the average molecular weight in the conclusions? This was not added that I could find. There is a distribution of molecular weights for any polymer, and they need to add this in.

Thanks the reviewer again. We have revised the conclusions. The molecular weight of polymers is commonly represented by the weight average molecular weight and the number average molecular weight, and the molecular weight distribution of polymers is the ratio of the weight average molecular weight to the number average molecular weight.

6They do not make good connections between the experiment and the calculated results. They need to make a stronger case in the text for this. They did not.

Thanks the reviewer. We have demonstrated from macroscopic experimental results that JXGZ demulsifier possesses excellent demulsification effect. Theoretical calculations reveal the demulsification mechanism of JXGZ at the atomic scale. The construction of molecular structure models for theoretical calculations is based on experimental data. The connection between experiments and theoretical calculations is already very clear.

7This paper still needs revision.

Many thanks for your great efforts on reviewing our paper and organizing the external review for this work. We have made multiple revisions to the entire text and believe that the revised version meets the normal requirements for publication in this journal
